# Novel Anti-Cancer Products Targeting AMPK: Natural Herbal Medicine against Breast Cancer

**DOI:** 10.3390/molecules28020740

**Published:** 2023-01-11

**Authors:** Bo Peng, Si-Yuan Zhang, Ka Iong Chan, Zhang-Feng Zhong, Yi-Tao Wang

**Affiliations:** Macao Centre for Research and Development in Chinese Medicine, State Key Laboratory of Quality Research in Chinese Medicine, Institute of Chinese Medical Sciences, University of Macau, Macao SAR 999078, China

**Keywords:** breast cancer, AMPK, natural products, metastasis, metabolism, immunity, multidrug resistance

## Abstract

Breast cancer is a common cancer in women worldwide. The existing clinical treatment strategies have been able to limit the progression of breast cancer and cancer metastasis, but abnormal metabolism, immunosuppression, and multidrug resistance involving multiple regulators remain the major challenges for the treatment of breast cancer. Adenosine 5′-monophosphate (AMP)-Activated Protein Kinase (AMPK) can regulate metabolic reprogramming and reverse the “Warburg effect” via multiple metabolic signaling pathways in breast cancer. Previous studies suggest that the activation of AMPK suppresses the growth and metastasis of breast cancer cells, as well as stimulating the responses of immune cells. However, some other reports claim that the development and poor prognosis of breast cancer are related to the overexpression and aberrant activation of AMPK. Thus, the role of AMPK in the progression of breast cancer is still controversial. In this review, we summarize the current understanding of AMPK, particularly the comprehensive bidirectional functions of AMPK in cancer progression; discuss the pharmacological activators of AMPK and some specific molecules, including the natural products (including berberine, curcumin, (−)-epigallocatechin-3-gallate, ginsenosides, and paclitaxel) that influence the efficacy of these activators in cancer therapy; and elaborate the role of AMPK as a potential therapeutic target for the treatment of breast cancer.

## 1. Introduction

Breast cancer is a common cancer in women worldwide. Notably in the United States, the estimated number of new cases was 287,850 and the death toll was 43,250 in women in 2022 [1]. Patients with breast cancer are diagnosed according to the activity of disease-associated markers, including human epidermal growth factor 2 (HER2) and hormone receptors (HR). Over 70% of breast cancer cases are characterized with an HR-positive and HER2-negative status [2]. Although breast cancer is divided into various pathological types, the status of estrogen receptor (ER), HR, and HER2, is an important basis for formulating treatment plans [3]. The highly positive correlations observed between female hormones and the development of breast cancer have encouraged researchers to concentrate on the discovery of targets that match breast cancer characteristics. Nowadays, surgical resection of local tumors is the principal choice for patients with nonmetastatic breast cancer. Other treatments, including chemotherapy, endocrine therapy, and radiotherapy, are also essential for the better management of breast cancer [4]. To improve clinical efficacy and limit vicious side effects during the course of treatment, targeted drugs that focus on pivotal proteins participating in the progression of breast cancer are in great demand. Several studies have demonstrated the essential roles of cell metabolism and energy generation in cell homeostasis. Breast cancer cells also need to rewire their metabolism to maintain cell survival, growth, and metastasis. Such a metabolism reprogramming process is known as the “Warburg effect” and is often accompanied by increased glucose uptake and lactic acid production [5]. Hence, the restriction and reversal of “Warburg effect” has been regarded as a promising strategy for the treatment of breast cancer.

Adenosine 5′-monophosphate (AMP)-Activated Protein Kinase (AMPK), a highly conserved serine/threonine kinase, acts as the central regulator of cellular metabolism and energy homeostasis in eukaryotic cells [6]. AMPK is involved in glucose metabolism, lipid production, protein synthesis, cell cycle regulation, immune responses, and many other biological activities through multi-level signaling networks. Generally, AMPK is sensitive to changes in the AMP (ADP)/ATP ratio and can be activated by liver kinase B1 (LKB1) or calcium-calmodulin-dependent protein kinase, kinase-β (CaMKKβ) [7,8]. Other drugs and biological metabolites, including metformin, doxorubicin, tamoxifen, 5-aminoimidazole-4-carboxamide1-β-D-ribofuranoside (AICAR), and oxidant hydrogen peroxide (H_2_O_2_) [9,10,11,12,13], as well as various stress conditions, such as hypoxia, hyperthermia, and glucose deprivation, can affect the activities of AMPK through specific mechanisms [14,15,16]. Activated AMPK shuts down the anabolic pathway of ATP consumption and promotes the control of glucose and lipid homeostasis to regenerate ATP [17,18]. Accumulating studies have confirmed that the activation of AMPK phosphorylation exerts multiple anti-cancer effects through initiating autophagy, inducing apoptosis, inhibiting proliferation, suppressing metastasis, stimulating the immune response, and reversing multidrug resistance (MDR) [19,20,21,22]. However, some studies demonstrated that the abnormal expression and activation of AMPK in cancer cells plays an important role in promoting proliferation, metastasis, and maintaining cancer cell survival under stress conditions [23,24,25]. A study investigating 92 pathway genes of AMPK in 21 types of cancers indicated opposite effects of AMPK on cancer progression [26]. Therefore, biomolecules in cancer cells and pharmacological modulators, including metformin, tamoxifen, as well as some other natural products, can manage cancer progression by regulating activities of AMPK and its downstream targets. The activation of AMPK regulated by pharmacological activators (such as metformin) is also involved in the regulation of type 2 diabetes, indicating that the regulation of AMPK activity directly interferes with the risk factors and progression of breast cancer [27]. In view of this, more efforts should be made to uncover the potential mechanisms and pharmacological modulators of AMPK, especially as a potential target in breast cancer, for the development of treatment strategies for breast cancer.

Our work has demonstrated that, in addition to synthetics, natural products from herbal medicines are potential sources in drug discovery for cancer treatment [28,29]. However, the use of AMPK as a target for natural products in the regulation of breast cancer progression have not been fully elucidated [30]. In this review, we systemically collected the characteristics and functions of AMPK and the associated signaling pathways of AMPK in breast cancer. The potential applications of AMPK in the treatment of breast cancer were analyzed with a focus on the expression, active status, and biological regulation of AMPK, as well as cell proliferation, cell death, cell metastasis, cancer metabolism, MDR, cancer immunity, and the tumor microenvironment. We also investigated synthesized chemicals and natural products that could activate AMPK to explore the pivotal role of AMPK in breast cancer management. Novel studies were reviewed from the PubMed, Web of Science, Medline, and Scopus databases to provide evidence to support AMPK as a potential therapeutic target for the treatment of breast cancer and to inspire novel target discoveries to manage breast cancer in the future.

## 2. Structure of AMPK

In eukaryotes, AMPK is a heterotrimeric complex that regulates cellular metabolism and energy homeostasis [31]. The structure of AMPK has been identified via electron microscope (EM) and X-ray crystallography, indicating the combination of a catalytic α-subunit, a scaffolding β-subunit, and a γ-subunit with regulatory abilities [32,33]. In mammals, each subunit has two or three subtypes, namely, α1α2, β1β2, and γ1γ2γ3, leading to 12 kinds of combinations of AMPKs [34]. AMPKs, which are encoded and expressed in different ways, exert a variety of physiological functions by changing their cellular locations. The α1, β1, and γ1 subunits are ubiquitous, whereas the α2 and β2 subunits are mainly expressed in skeletal and heart muscles. γ2 is expressed in the heart and a few other tissues, whereas γ3 is merely expressed in skeletal muscles [35].

Typically, the α1 and α2 subunits contain 559 amino acids and 552 amino acids, respectively, which are composed of a kinase domain (KD), an autoinhibitory domain (AID), an α-linker, and a C-terminal domain (α-CTD), in that order [36]. KD lies at the N-terminal of the α-subunit and has an activation loop with a Thr172 (Thr174 in human α1) phosphorylation site for the activation of AMPK [31,37]. KD functionally interacts with the C-terminal of the β subunit and cystathionine β-synthase (CBS) is repeated in the γ subunit [36]. The AID domain contains three α-helices and is stabilized by hydrophobic residues [38]. In addition, AID interacts with the lobes of KD, which keeps AMPK inactive [33]. An α-regulatory subunit-interacting motif (α-RIM) is present in the α-linker, which is an intermediary in the allosteric activation of AMPK. The C-terminal of the α subunit combines with the β subunit and the γ subunit to sustain the heterotrimeric structure.

In humans, AMPKβ1 is a protein consisting of 270 amino acids and AMPKβ2 contains 272 amino acids [39]. The β subunit contains a myristoylation site at the N-terminal, followed by a carbohydrate-binding module (CBM), a β-linker loop, and β-CTD, which acts as a scaffold, similarly to α-CTD [36]. The myristoylation site transfers proteins to the plasma membrane and endomembrane surfaces [40]. CBM is sensitive to glycogen, which leads to the inactivation of AMPK, which is manifested by the disruption of the interactions between CBM and KD (Figure 1). The interface between KD and CBM offers a unique site for pharmacological activators of AMPK [41].

There are three isoforms of the γ subunit in humans. The γ1 subunit has 331 amino acids, γ2 has 569 amino acids, and γ3 consists of 489 amino acids. The γ subunit has two symmetrical Bateman domains, which contain four CBS-associated sites in total. CBS sites competitively bind to AMP, ADP, and ATP and consequently transfer allosteric modifications [31]. Site 2, which lacks the key aspartate that is essential in nucleotide binding, fails to regulate AMPK activity, whereas other sites, including sites 1, 3, and 4, have different affinities for adenosine phosphates. When AMP, or sometimes ADP, binds to site 3 of the γ subunit, α-RIM is stimulated to wrap around adenosine, modifications of which induce the exposure of KD from AID and initiate the allosteric activation of AMPK.

## 3. Aberrant Expression of AMPK in Breast Cancer

The expression of AMPK in breast cancer cells is usually different from that in normal breast cells. Bioinformatics analysis has demonstrated a relationship between downregulated differentially expressed genes (DEGs) and the AMPK signaling pathway [42]. In one study, the authors found that the AMPK was positively expressed in epithelial cells by comparing tissue samples from 449 breast cancer patients with 27 normal breast and fibroadenoma tissue samples, and the positive expression of AMPK was associated with recurrence rate, lymph node involvement and poor survival rate [43]. The overexpression of AMPK is partially attributed to the loss-of-function mutation of microRNA-101-3p, which targets the 3′-untranslated region (3′-UTR) site of AMPKα1 mRNA and inhibits the expression of AMPK [23]. In addition, AMPK protein levels have been positively correlated with the overexpression of lactate dehydrogenase A (LDHA) in triple-negative breast cancer (TNBC) cells and breast cancer tissues, and this has been attributed to increased glucose intake and efficient glycolysis. Patients with high levels of AMPK and LDHA usually exhibit poor Tumor Node Metastasis (TNM) stages, distant metastasis of tumors, and accelerated cancer progression with Ki67-positive features [23,44].

By contrast, an immunohistochemical analysis of AMPKα1 expression profiles indicated a negative correlation between AMPKα1 expression and human mammary cancer metastasis and poor prognosis. Mechanistically, frequent mutation of phosphatidylinositol-4,5-bisphosphate 3-kinase (PI3K), and overexpression of HER2 in breast cancer downregulated the ΔNp63α (a predominant p63 isoform) which disrupted the transcription and expression of AMPKα1 and subsequently inhibited the expression of E-cadherin [45]. MicroRNA 27a was highly expressed in MCF-7 cells and repressed the AMPKα2 expression by binding to the 3′-UTR site of AMPKα2 mRNA [46]. AMPKα2 is qualified to manage the mammalian target of rapamycin (mTOR) signaling pathway to control cancer cell cycle arrest. Decreased cyclin D1 and increased nuclear p53 protein levels, which enhanced cancer apoptotic events, were observed in AMPKα2 xenograft models [47]. Therefore, the existing studies on the expression of AMPK in breast cancer is still controversial.

## 4. Abnormal States of AMPK in Breast Cancer

The expression of the AMPK protein varies in different cell types. In general, activity of the AMPK is inhibited in most breast cancer cell lines due to the mutation of p53. The p53 protein, a molecular tumor suppressor, targets the Sestrin 2 which forms a complex with AMPK and tuberous sclerosis complex 2 (TSC1)-tuberous sclerosis complex 2 (TSC2). In this complex, Sestrin 2 interacts with the AMPKα subunit and stimulates AMPK activation. The activation of AMPK phosphorylates TSC2, of which phosphorylation further modulates the activity of mTORC1 [48]. The mTORC1 complex, composed of mTOR, raptor, PRAS40 and mLST8, is responsible for the regulation of cell growth and protein synthesis, and is considered to be one of the important parts of the mTOR signaling pathway [49]. The accumulation of gene mutations accelerates the breast cancer development. Mutant p53 proteins prevent the formation of autophagic vesicles in breast cancer cells and inhibit the phosphorylation of AMPK at Thr172 [50]. Phosphatidylinositol-4,5-bisphosphate 3-kinase catalytic subunit alpha (*PIK3CA*), encoding the p110α catalytic subunit of PI3K, as well as phosphatase and tensin homolog (PTEN), which are the suppressors of the PI3K/protein kinase B (Akt) signaling pathway, are frequently mutated in breast cancer [51,52]. The activation of Akt phosphorylates AMPKα1 at Ser485 (Ser487 in human α1) and leads to subsequent inhibition of AMPKα1 activity by dephosphorylating Thr172 [53]. However, treatment with AICAR, an AMPK activator, causes the inhibition of Akt phosphorylation at Ser473 in breast cancer cells, suggesting the AMPK inversely regulates the activity of Akt [54]. The double-negative feedback loop between AMPK and Akt interactively influence the breast cancer metastasis. The activities of AMPK and Akt show opposite states when breast cancer cells are in matrix-attached and matrix-detached conditions. Once the cancer cells are in a matrix-detached condition, AMPK is activated in order to suppress Akt phosphorylation and to resist anoikis [55]. However, the authors of another study observed the Akt1 and Akt2, two of three isoforms of Akt, promoted TNBC cells to overcome apoptosis through reducing Bim expression or upregulating Mcl-1 expression, and AMPKα1 and AMPKα2 have limited effects in this process [56]. Therefore, whether the interaction between AMPK and Akt mediates the breast cancer cell metastasis still needs further study.

Breast cancer mostly occurs in women and estrogen levels have an impact on the activity of AMPK. Estrogen receptor alpha (ERα)-driven signals are important in luminal breast cancer in the regulation of the p53/AMPK axis. ERα can bind to the p53 protein and antagonize wild type p53 protein to repress AMPK phosphorylation [57,58]. It is worth noting that the inhibition of AMPK is also observed, in the absence of ERα, in breast cancer cells harboring a mutant p53 protein, indicating ERα and mutant p53 proteins competitively inhibit AMPK phosphorylation [59]. In addition, ERα and estrogen receptor beta (ERβ) are able to interact with the γβ-bind domain of AMPKα2 subunit directly, and ERα activation triggered by 17-β oestradiol (E2), a main circulating estrogen, is essential for E2-dependent AMPK activation. Moreover, if the activity of ERβ is inhibited, the E2-induced activation of AMPK phosphorylation can be enhanced in breast cancer [60].

The development of cancer is often accompanied by increased pressure and interstitial fluid flow (IFF). The shear stress induced by IFF can activate AMPK by the activation focal adhesion kinase (FAK) and Src, and the effect of IFF on AMPK activation is influenced by the subcellular location of AMPK, indicating that the physical environment of a tumor also influences the activity of AMPK [61,62]. The secretion of matricellular proteins is one of the characteristics of the tumor microenvironment. Cysteine-rich angiogenic inducer 61 (CYR61), a matricellular protein of the CCN family, is highly expressed in TNBC cell lines and promotes lung metastasis of MDA-MB-231 cells by activating AMPK [63,64]. Cardiotrophin 1 (CTF1) derived from cancer cells was reported to induce protective autophagy by activating AMPK in fibroblasts from tumor stroma [65]. Other biomolecules exist in breast cancer cells, such as the overexpression of cyclin D1 which phosphorylates LKB1 at Ser325, reversing the autophagy induced by AMPK activation under stress conditions [66]. Overexpressed ubiquitin-conjugating enzyme E2O (UBE2O) in human cancer induces the ubiquitination of AMPKα2 at K470, which further promotes breast cancer progression and metabolic reprogramming [67]. These existing studies demonstrate that the activities of AMPK are influenced by gene mutations, estrogen changes and tumor microenvironment in breast cancer cells.

## 5. Pleiotropic Regulations of AMPK in Breast Cancer

### 5.1. Regulation of Breast Cancer Cell Proliferation by Targeting AMPK

Interrupting the cell cycle and inhibiting biosynthesis are important strategies to inhibit the growth and proliferation of cancer cells. Polycomb repressive complex 2 (PRC2), composed of Suz12, Eed, Enhancer of zeste homolog 1/2 (Ezh1/2) and RbAp46/48, plays a vital role in maintaining stem cell pluripotency and in promoting cancer cell proliferation [68]. AMPK phosphorylates Ezh2, a necessary histone methyltransferase, at T311 site, and in turn disrupts the interactions between Ezh2 and Suz12, which leads to the inhibition of tumor growth [69]. HER2 and epidermal growth factor receptor (EGFR) overexpressed in some breast cancer cell lines promote the growth and proliferation of breast cancer cells. The activation of AMPK phosphorylation inhibits the activity of HER2 and EGFR, which further suppresses the growth of breast cancer [70]. Disheveled segment polarity protein 3 (DVL3), an upstream modulator of the Wnt/β-catenin signaling pathway, significantly promotes breast cancer progression. Upon treatment with metformin, an AMPK activator, the levels of DVL3 and β-catenin were decreased in MCF-7 and MDA-MB-231 cells, and the transcription of two downstream molecules of β-catenin, c-MYC, and cyclin D1, were repressed [71]. It has been reported that the knockdown of glycogen synthase kinase-3β (GSK-3β) enhanced the activation of AMPK phosphorylation [72], whereas the activation of GSK-3β and sirtuin 1 (SIRT1) induced by AMPK were found to be responsible for the inhibition of c-MYC and metadherin expression in TNBC oncogenesis [73]. AMPK is a modulator of TAZ and YAP [74]. As key molecules of the Hippo signaling pathway, YAP and TAZ are responsible for regulating the expression of genes related to cell growth, and their activities are related to their positions in the cell [75]. Potent modulators, such as PLD6, mediate the MYC-induced mitochondrial fusion and induce AMPK activation which further inhibits the activities of YAP/TAZ [76]. Mitosis defects are a hallmark of tumorigenesis. Flexible control of the cell cycle by means of mitosis checkpoint-associated modulators, such as CDK, can sustain the fidelity of cell proliferation. CDK1 regulates chromosomal alignment and mitosis through phosphorylating the α1, α2, and β1 subunits of AMPK [77]. CDK inhibitors, such as p21 Waf1/Cip1, therefore exhibit particular potential in the clinical management of advanced breast cancer [78,79].

### 5.2. Regulation of AMPK: Promoting Breast Cancer Cell Death

Although many studies have shown that the AMPK activation can induce protective autophagy to avoid cancer cell apoptosis under stress conditions, some downstream molecules of AMPK involved in the apoptosis induction caught our attention [7,24]. Apoptosis is generally induced by AMPK activators and exhibits different manifestations that are dependent on the cell types [80]. Apoptosis can be rapidly induced by tumor necrosis factor (TNF)-related apoptosis inducing ligand (TRAIL), a member of the TNF superfamily, which binds to death receptor 4 (DR4) on the cell surface and promotes the expression of CCAAT-enhancer-binding protein homologous protein (CHOP) and Bax, activation of caspase 3 and caspase 9, as well as reduces the level of B-cell lymphoma-2 (Bcl-2) protein in an AMPK activation-dependent manner [81]. The enhancement of the expression of death receptor 5 (DR5) partially promotes the breast cancer sensitivity to TRAIL for AMPK-related apoptosis, demonstrating an advantage in combined chemotherapies [82]. When breast cancer cells undergo apoptosis, canonical apoptotic bioregulators, involving the cleavage of caspases, the translocation of Bax to the outer mitochondria membrane, the release of cytochrome-c, and the activation of poly ADP-ribose polymerase (PARP), occur in an orderly manner, which favors AMPK activation [80,83]. Bim, one member of the pro-apoptotic Bcl-2 family, can be upregulated by the pharmacological activators of AMPK. AMPK activity exhibits a positive correlation with Bim levels and co-administration of metformin with Bcl-2/B-cell lymphoma-XL (Bcl-XL) inhibitors can promote the apoptosis of MYC-driven breast cancer cells [19].

In addition, inflammasomes, the representative indicators of pyroptosis, exhibits bidirectional effects on cancer progression. Pyroptosis-associated proteins, including NOD-like receptor pyrin domain-containing protein 3 (NLRP3), the apoptosis-associated speck-like protein containing a CARD (ASC), and caspase-1, are overexpressed and are abnormally activated in most cancer types, especially in breast cancer [84]. Controversially, certain inflammasome inducers with the ability to initiate cancer cell pyroptosis are regarded as potent anti-cancer treatments [85]. This phenomenon, along with accumulating evidence on crucial roles of inflammasome components in the oncogenesis of AMPK-related breast cancer, has emphasized the importance of the nature of chemical molecules and the contexts of specific cancers [86]. The activation of AMPK/SIRT1 triggers mitophagy and pyroptosis induced by gasdermin E-N (GSDME-N) [87]. Another study has indicated AMPK-drives pyroptosis, with necessary contributions from breast cancer susceptibility gene 1 (*BRCA1*), which is frequently mutated in hereditary breast cancer [88].

Autophagy is considered a self-protective mechanism, which protects cells against damaged organelles. Accumulating studies have demonstrated the elusive roles of autophagy in breast cancer oncogenesis. In the autophagy-associated network, the AMPK/mTOR axis is the crucial factor that responded to deviant energy circulation, activation of caspase-9 and PARP, and reduced expression of Bcl-2 and other biological functions under severe stress conditions [89]. Both AMPK and mTOR activation can directly bind to Unc-51-like kinase 1 (ULK1) to modulate cellular autophagy in contrasting ways [90]. The autophagy stimulated by icaritin via the activation of AMPK and ULK1 partially contributes to the inhibition of breast cancer proliferation. On the other hand, additional treatment with the autophagy inhibitor could promote these inhibitory effects, thus demonstrating the directional functions of autophagy for cellular self-protection in breast cancer [91]. The inhibition of bromodomain-containing protein 4 (BRD4), which acts as an epigenetic memory modulator, could lead to autophagy protein 5 (ATG5)-dependent autophagy-related cell death via interaction with the AMPK/mTOR/ULK1 axis [92]. In addition, the authors of a recent study observed the involvement of the AMPK/SIRT1/p53 signaling pathway in nitrosative stress-induced autophagy-associated cell death [93].

As for other phenotypes of cell death, there remains a small body of research focusing on determining the precise role of AMPK in breast cancer. Ferroptosis, an iron-dependent programmed phenotype, is considered a novel target in breast cancer, attributed to its involvement in the redox regulations of reactive lipid hydroperoxides. Both activation and inhibition of AMPK has been closely related with ferroptosis involved in stress responses. Interestingly, metformin promotes ferroptosis processes in an AMPK-independent manner, indicating that there are parallel molecules in AMPK-associated signaling pathways possessing undetermined regulatory features [94]. The precise mechanisms of action of the discovered regulators need to be determined and potent participants in breast cell ferroptosis need to be identified for use in anti-cancer treatments.

### 5.3. The Central Role of AMPK in Breast Cancer Stem Cells (BCSCs), as Well as Metastasis and Angiogenesis

BCSCs are a group of cells characterized by self-renewal, tumorigenicity, and low differentiation. Generally, BCSCs are highly heterogeneous, which leads to different phenotypes and functions of breast cancer cells in the same tumor. Although the number of BCSCs only accounts for a small part of the total number of breast cancer cells, the presence and state of BCSCs are highly correlated with metastasis and recurrence [95]. It has been demonstrated that two common phenotypes of BCSCs, CD44(+)CD24(-low)Lineage(-) and aldehyde dehydrogenase 1 + (ALDH1+), are associated with breast cancer metastasis and poor prognosis [96,97]. BCSCs are also affected by various cytokines and the tumor microenvironment. For instance, the overexpression of multiple copies in T-cell malignancy 1 (MCT-1) stimulated the secretion of IL-6 and regulated the receptor levels of IL-6, which enhanced the stemness of BCSCs [98]. Furthermore, IL-8 also played an important role in the self-renewal of BCSCs, and a study found that the blockage of IL-8 receptor CXCR1 inhibited FAK/Akt signaling pathway and activated forkhead box O transcription factor 3a (FOXO3a), which led to a decrease in the number of BCSCs [99]. The activation of GSK-3β was also considered to be related to the mesenchymal properties of BCSCs, which was confirmed by the strong migration ability of TNBC cells [100]. In addition, the phenotype of BCSCs exhibited a positive relationship with the level of P-glycoprotein (P-gp), which was one of the reasons why BCSCs can overcome drug resistance and lead to breast cancer recurrence [101]. These cases illustrated that the inhibition of breast cancer metastasis, drug resistance, and recurrence by modulating BCSCs activity will have huge benefits and these molecules can be mediated by AMPK. Metformin has exhibited stronger cytotoxicity against BCSCs, mainly through the activation of the AMPK/mTOR axis [15,102]. In addition, metformin promoted glucose consumption and lactate production in BCSCs, which further impaired the ability of BCSCs to repair DNA damage [103]. The deficiency of LKB1, an upstream molecule of AMPK, was associated with the acquisition of the phenotypes of BCSCs. Honokiol inhibited the phosphorylation of signaling transducers and activators of transcription 3 (STAT3) activation in an LKB1/AMPK-dependent manner, which prevented STAT3 recruitment and suppressed the expression of iPSC inducers (including Nanog, Oct4 and Sox2) [104]. The overexpression of UBE2O in breast cancer cells significantly promoted the proliferation, epithelial-mesenchymal transition (EMT) and acquisition of stem cell characteristics. This effect may be related to the ubiquitination and degradation of AMPKα2 [105]. However, a recent study inversely shown that AMPK promoted the mammosphere formation and the maintenance of BCSCs. In this process, AMPK promoted the expression of *SOX2*, *BMI1* (Appendix A) and *NANOG* genes which belong to the stemness gene through the transcriptional upregulation of twist family bHLH transcription factor 1 (TWIST1) [106]. Furthermore, the transition of BCSCs between epithelial-like and mesenchymal-like states keeps BCSCs in balance between proliferation and invasiveness. AMPK activation regulates the transition from the invasive mesenchymal-like state to the proliferative epithelial-like state in BCSCs via promoting the expression of hypoxia-inducible factor-1α (HIF-1α) protein [107].

Cell metastasis is a complicated process which gradually leads to cancer propagation. The cancer cells are forced to lose epithelial-like features and invade the body through blood vessels, spreading into distant organs, a process which is associated with drug resistance and disease recurrence [108]. Angiogenesis is obviously an active part of tumor development, sustaining cell survival, which requires sufficient energy to be derived from aerobic glycolysis. Nuclear factor erythroid 2-related factor 2 (Nrf2), the key responder to aberrant oxidative stress, can be mediated by vascular endothelial growth factor (VEGF)/extracellular signal-regulated protein kinase (ERK) 1/2 axis [109]. Under hypoxia, HIF-1α which is enhanced upstream by Nrf2, accompanied by Akt activation and AMPK repression, is required for this regulatory pathway [110]. Interestingly, CD44 ablation could induce enhanced AMPK and suppressed HIF-1α and LDHA activity, as CD44 knockdown results in a reduction of c-Src and Akt activity [111]. In another study investigating transforming growth factor-β1 (TGF-β1)-induced EMT in MCF-7, both metabolic reprogramming and oxidative rebalance were observed, accompanied by the phosphorylation of AMPK [25]. Interestingly, integrins on the cell surfaces of both cancer cells and cancer-associated fibroblasts (CAFs), which are adopted in fibronectin-dependent adhesion, could be suppressed upon AMPK activation, coupled with mitophagy-related characteristics [112,113]. Although AMPK activation is considered to maintain cell survival, AMPK activation can inhibit cell metastasis and distant adhesion by reducing β1-integrin on the surface of cell membrane [112]. However, another study showed that β4-integrin overexpressed CAFs induced the overexpression of TWIST1, snail family transcriptional repressor 2 (SNAI2), zinc finger E-box binding homeobox 1 (ZEB1), ZEB2, and N-cadherin by activating AMPK phosphorylation, leading to mitophagy and lactate production in CAFs [113].

Although many studies have demonstrated that activated AMPK can inhibit the metastasis of breast cancer, it is worth exploring how abnormally activated AMPK plays a negative role in metastatic breast cancer. In the process of tumor invasion, the activation of AMPK phosphorylation provides sufficient energy through glucose metabolism and the regulation of ATP/ADP homeostasis, enhancing the lifetimes of leader cells [114]. When circulating breast cancer cells are attached to distant locations, the cytoskeletal network, comprising microtubules and cofilin, is concurrently inhibited by AMPK activation [115]. Furthermore, activated AMPK directly drives the phosphorylation of the catalytic alpha subunit of the pyruvate dehydrogenase complex (PDHc) (PDHA) to sustain the tricarboxylic acid (TCA) cycle, ultimately resisting metabolic and oxidative stress at colonized locations [116,117]. During metastatic colonization, breast cancer cells are required to resist cell deathinduced by extracellular matrix deprivation, which is termed anoikis. AMPK is initiated to suppress Akt with the help of pleckstrin homology domain leucine-rich repeat protein phosphatases 2 (PHLPP2) to combat anoikis [55]. As an important part of cancer cell metastasis, AMPK activation can be induced by EPH receptor A2 (EPHA2)-enriched exosomes via a canonical forward signaling pathway that is dependent upon the ligand Ephrin A1, indicating that EPHA2, Ephrin A1, and HIF-1α, which are specifically contained in the tumor microenvironment, jointly promote AMPK activation and angiogenesis [118]. In addition, strong AMPK activation has been found to promote the detachment and anoikis resistance of MDA-MB-231 cells [119]. Together with phosphorylation at Ser166 of the phosphoprotein, which is enriched in astrocytes 15 (PEA15), the activation of AMPK can enable human mammary epithelial cells to survive in suspension, in turn leading to the anchorage-independent growth of breast cancer cells [120].

### 5.4. Metabolism of Breast Cancer Is Regulated by AMPK

The Warburg effect, characterized as aerobic glycolysis, generally occurs in tumors, ensuring abundant energy and nutrient availability for cell proliferation and cancer metastasis. Therefore, the rewiring of the cell metabolism, especially of glucose, seems like a potent strategy for the targeting of breast cancer. AMPK is involved in the expression of glucose transporters (GLUTs), such as GLUT1, which render the channels for glucose uptake [121]. Manganese superoxide dismutase (MnSOD) is a mitochondria-resident enzyme that transforms mild oxidant superoxide radicals into strong H_2_O_2_. The regulation of ATP production and other mitochondria-driven energetic functions by MnSOD contributes to aggressive cancerous effects, which is mediated by AMPK activation and maintenance of glycolysis [122]. In addition to augmented glycolysis, tumor metabolism is characterized by disrupted lipid metabolism, aberrant cholesterol, and protein synthesis [123]. The transcriptional expression of proline dehydrogenase (POX) induced by AMPK activation regulates proline metabolism, which also promotes protective autophagy [124]. AMPK cooperates with biological modulators to respond to the abundant consumptions of glucose, or glutamine as an alternative. FOXO3a is promoted by glucose stress-induced AMPK activation and FOXO3a dramatically upregulates hematopoietic PBX1-interacting protein (HPIP) to increase glutamine uptake in response to glucose stress. Interestingly, prolonged glucose stress leads to degradation of HPIP, leading to cell death in order [125]. AMPK activation reprograms folate cycle metabolism, related to the feature of purine biosynthesis, which is restrained by the proliferator-activated receptor γ co-activator 1α (PGC-1α)/estrogen-related receptor α (ERRα) axis [126]. In addition, AMPK mediates lipid metabolism by targeting acetyl-CoA carboxylases (ACCs), whereby the phosphorylation of ACC1 at Ser79 and ACC2 at Ser212 by AMPK results in the suppression of fatty acid synthase (FASN) [127]. The inactivation of ACC1 by AMPK may also be triggered by the inhibition of 6-phosphogluconate dehydrogenase (6PGD) [128]. Generally, cancer cells utilize de novo lipogenesis to satisfy their rapid growth, which requires sufficient lipids for membrane integrity and the biosynthesis of signaling modulators [129]. The depletion of O-linked β-N-acetylglucosamine transferase (OGT) promotes the activation of AMPK via phosphorylation at Thr172, and the activation of AMPK subsequently enhances sterol regulatory element-binding protein 1 (SREBP1) phosphorylation at Ser372, finally impeding FASN and breast cancer growth [130,131]. Moreover, fatty acid oxidation (FAO) for energy supplementation modulated by carnitine palmitoyltransferase-1 (CPT-1) as well as CD36 is enhanced in an AMPK-dependent manner, especially during cancerous metastatic processes [25]. Other biomolecules, such as MYC and HER2, have been observed to process oncogenesis by supporting FAO and fatty acid uptake within a complicated signaling network [132,133].

### 5.5. AMPK and Multi-Drug Resistance in Breast Cancer

Acquired MDR is the main reason for the failure of chemotherapy and radiotherapy. The cancer cells with MDR usually exhibit characteristics such as apoptosis resistance, alternative metabolic system, and aberrant transporters [22]. Similar to the effect of AMPK in promoting metastasis, the negative effect of AMPK activation was also observed in breast cancer cells with MDR. Part of this resistance to chemotherapeutics and targeted therapy may be triggered by ROS- or estrogen-induced changes in AMPK activity [134,135]. For instance, drug resistance during tamoxifen treatment may be related to the increased expression of metastasis-associated 1 (MTA1) induced by tamoxifen. Highly expressed MTA1-induced protective autophagy is associated with AMPK activation [136]. Similar to the MTA1-associated tamoxifen resistance, the AMPK-triggered MDR is also unique to doxorubicin-resistant breast cancer cells sustained by protective autophagy and activated ULK1 [137]. In addition, the positive correlation between the expression of the transient receptor potential channel 5 (TRPC5) and autophagic states is associated with the CaMKKβ/AMPKα/mTOR/p70S6K signaling pathways in adriamycin-resistant breast cancer cells [138]. Molecules targeting Ca^2+^ channels, such as transient receptor potential melastatin 8 (TRPM8), which interacts with AMPK and activates the AMPK/ULK1 signaling pathway, might be potential regulators of oncogenesis [139]. Even in radiotherapy, the activation of AMPK induced by dopamine receptor D2 activation restrains the cleavage of PARP, which is correlated with the inhibition of the radiosensitizing effect of aripiprazole [140]. Therefore, the combination of first-class chemotherapies with potent chemosensitizers, such as AMPK regulators, could reverse autophagy-induced MDR and impede cancer progression [83,141].

The overexpression of ATP binding cassette (ABC) transporters is an important reason for the reduction of drug accumulation in cells, which can be reversed by the activation of AMPK. Another issue associated with MDR is the overexpression of P-gp, which reduces accumulation of drugs, preventing them from having sufficient efficacy, and is downregulated by AMPK/HIF-1α axis activation [142]. Arachidonate lipoxygenase 12 (Alox12), which transforms arachidonic acid into 12-hydroxyeicosatetraenoic acid (12-HETE) instigates breast cancer cells to resist the cytotoxicity of paclitaxel or 5-flurouracil [143]. The deletion of Alox12 contributes to the restoration of chemosensitivity by activating AMPK and restraining ACC1-associated lipid biosynthesis [143]. Inhibition of AMPK activity also leads to distant metastasis of breast cancer cells and is the main reason for the failure of chemotherapy. For example, the AMPK/mTOR axis activated by the knockdown of mesenteric estrogen-dependent adipogenesis gene (*MEDAG*) further promotes the expression of E-cadherin and inhibits the expression of N-cadherin and Snail, restoring epirubicin sensitivity [144].

Although many studies have demonstrated that AMPK activation is associated with autophagic protection-induced treatment failure, other studies claimed that AMPK activation can augment the effect of antitumor drugs by regulating ABC transporters, biosynthesis, and cell adhesion (Figure 2).

### 5.6. Cancer Immunity: A New Target for AMPK

With the huge successes in the development of immune pharmaceuticals, particularly employing immune checkpoints or adoptive cell therapies, the connections between breast cancer and its immune microenvironment have attracted great interest. Programmed death ligand 1 (PD-L1) is located on the cancer cell membrane and in the cytoplasm, and can be monitored by T cells, consequently regulating cancer cell escape [145]. Approaches to suppressing PD-L1 are considered potent cancer treatments, with exciting outcomes. AMPK phosphorylates the Ser195 site of PD-L1, leading to PD-L1 glycosylation and degradation [146]. Some downstream molecules, including D-mannose, as well as the expression of histone deacetylase (HDAC) proteins, were reported to facilitate immunotherapies-an effect which was attributed to PD-L1 degradation, which is dependent on AMPK activation [147,148]. In addition to acting on PD-L1 directly, AMPK activation augments the effect of PD-1 blockade via promoting PGC-1α expression [149].

Among immune infiltration cells in cancer, macrophages are the most abundant component, and their metabolic status can be dynamically altered to sustain regulatory functions. Tumor-associated macrophages (TAMs) are a polarized M2-subtype that support cancer progression. Chemokines, especially CC chemokine ligand 5 (CCL5), secreted by lactic acid-stimulated TAMs, were able to interact with CCR5, which activated AMPK-associated autophagy [150]. Under hypoxia condition, AMPK activation mediates the expression and secretion of galectin-3 in TAMs, but this positive regulation of galectin-3 by AMPK was inhibited in the presence of metformin, illustrating the complexity of TAMs and AMPK regulatory pathways [151]. Hence, this positive metabolic feedback loop comprising TAMs and cancer cells using glycolysis implies the complicated nature of cancer immune systems. T cells are another participant in immune surveillance, and they exert cytotoxicity effects in a dynamic manner. Natural killer (NK) cells can induce cancer cell lysis partly by the released pro-apoptotic granules. Granzyme B (GzmB) is one principal effector of NK cells and supports NK cell-induced elimination of breast cancer established with the reactivation of p53. Mechanically, this anti-cancer potentiation is attributed to the induction of autophagy via the Sestrin/AMPK/ULK/mTOR axis [152]. In conclusion, AMPK plays a pivotal role in both the cancer and the immune cell contexts to meet their metabolic requirements in the feedback loops. More insights can be gained in relation to other components, such as B cells and regulatory T cells, which have not been investigated in terms of their specific relationships with AMPK-associated immune monitoring in breast cancers.

### 5.7. Various Molecules Regulated by AMPK in the Tumor Microenvironment

Some biomolecules secreted from cancer cells jointly create a microenvironment for tumor growth and metastasis. Myeloid-derived suppressor cells (MDSCs) are a pivotal component of the immunosuppression process that restrains cytotoxic T lymphocytes (CTLs) and enhances tumor survival. The activation of liver-enriched activator protein (LAP), an isoform of CCAAT/enhancer-binding protein beta (CEBPB), as well as the expression of granulocyte colony-stimulating factor (G-CSF) and granulocyte macrophage colony-stimulating factor (GM-CSF), were reported to affect the development of MDSCs in TNBC cell lines. This process is initiated by the restriction of glycolysis and the associated induction of AMPK/ULK1 and autophagy pathways [153]. This suggests the potential role of glycolysis restraint induced by LDHA knockdown in suppressing MDSCs and, consequently, in enhancing T cell immunity, especially in TNBC. Prostaglandin 2 (PGE_2_) is significantly secreted from doxorubicin-resistant breast cancer cells and promotes the expansion of MDSCs via the microRNA 10a/AMPK axis, which is significantly different from the presented above [154]. More importantly, feedback regulation has been observed between microRNA 10a, AMPK, MDSCs and protein kinase A (PKA). The activation and expansion of MDSCs activates PKA, which in turn promotes the generation of microRNA [154].

Bio-indicators from certain diseases that are correlated with oncogenesis are an essential part of treatments targeting the cancerous microenvironment. Leptin is a pleiotropic hormone produced by adipose tissues which diminishes the anti-proliferation effects of AICAR through the inhibition of AMPK phosphorylation [54]. The pro-oncogenic role of leptin is localized in the ER/AMPK axis, coupled with the induction of cellular autophagy [155]. Adiponectin is a glycoprotein belonging to the adipokines and the pro-apoptotic effects of adiponectin have been comparatively observed in ERα-negative breast cancer cells, with results indicating the pivotal roles of ER-driven AMPK and other modulators, such as FOXO3a and LKB1 [156,157,158]. Furthermore, globular adiponectin causes the integral suppression of endoplasmic stress and the NLRP3 inflammasome in restrained breast cancer cells [86].

In addition to the biomolecules secreted by cancer cells, the specific nutritional conditions that support tumor growth also affect the survival of all cancer cells [159,160]. In low D-glucose conditions, AMPK and G protein-coupled estrogen receptor 1 (GPER1) are found to be constitutively accumulated in MCF-7 cells [161]. AMPK mediates the demethylation of H3K36me2 via lysine-demethylase 2A (KDM2A) in response to mild glucose starvation and reduces rRNA transcription [162]. Hypoxia is a special hallmark of solid cancer [163]. The alternations caused by hypoxia result in the enhancement of EMT, angiogenesis, and the acquisition of MDR, in which HIF-1α is the dominant player. In the hypoxia-associated modulator network, the upstream AMPK targets HIFs and mTOR, subsequently initiating the transcription of HIFs target genes in response to hypoxia [164]. Simultaneously, chromatin structure modifications and ribosomal translocations have been observed in hypoxia-stressed breast cancer cells [164,165].

## 6. Potential AMPK Modulators of Chemical Synthesis

### 6.1. 5-Aminoimidazole-4-Carboxamide1-β-D-Ribofuranoside (AICAR)

AICAR is an analog of AMP and is used to directly activate AMPK in pharmacological experiments. On the one hand, AICAR attenuates “Warburg” metabolism and improves the immune responses of T cells and B cells to enhance their synergistic effects with other anti-cancer agents [166,167]. In a study employing prostate cancer cells, AICAR phosphorylated and inactivated ACC, augmented the X-rays-induced clonogenic killing capacity and spheroid growth impedance in a p53-independent way [168]. Interestingly, the co-treatment with AICAR and methotrexate inhibited MCF-7 cell proliferation but had little influence on cell death. AICAR and methotrexate elevated mitochondrial oxidation and decreased glycolysis in this process [169]. 6-phosphofructo-2-kinase/fructose-2,6-bisphosphatase isoform 3 (PFKFB3) is an upstream molecule that is also regulated by AMPK. Knockout of *PFKFB3* inhibits the activation of AMPK and consequently promotes apoptotic events. The administration of AICAR was not able to reverse the cytotoxicity induced by PFK-15, an inhibitor of PFKFB3, even though AICAR rescued the activity of AMPK. Obviously, as a commonly used synthetic activator of AMPK, the antitumor effects of AICAR are partially exerted in an AMPK-independent manner [170].

### 6.2. Doxorubicin

Doxorubicin is a broad-spectrum anti-cancer agent which is recommended by the National Comprehensive Cancer Network (NCCN) for the treatment of HER2-negative breast cancer, but the resistance of doxorubicin becomes an obstacle to obtaining satisfactory efficacy in clinical practice. Mechanistically, doxorubicin embeds in DNA segments and obstructs topoisomerase II, which repairs damaged DNA [171]. Treatment with doxorubicin triggers autophagic processes, including activating autophagic flux, increasing the LC3B-II/I ratio, decreasing p62 expression, and forming autophagosomes, in MCF-7 cells. Previous studies suggested that blockage of AMPK reversed these autophagic processes and promoted doxorubicin-induced apoptosis [137]. On the contrary, in mice with solid Ehrlich carcinoma, an impedance in AMPK phosphorylation was observed after the administration of doxorubicin, which indicated that AMPK was able to play opposing roles that are dependent on the cell context [171]. Concomitant administration of doxorubicin and metformin potently enhanced antitumor efficacy by activating AMPK [171]. A study indicated that SIRT1 and AMPK were inhibited in non-small cell lung carcinoma (NSCLS) cells when exposed to hypoxia, which conferred doxorubicin resistance in solid tumors [172]. Thus, activating AMPK in case of environmental stress is a possible means of restoring doxorubicin sensitivity, as suggested in a study with U2-os osteosarcoma cells [173]. In human hepatocellular carcinoma BEL7402 and SMMC7721 cells, the administration of doxorubicin strongly promoted the expression and translocation of high mobility group box protein 1 (HMGB1) which further triggered protective autophagy and impeded apoptosis induction via activation of AMPK/mTOR signaling pathway [174]. These studies showed that AMPK is involved in the autophagy or apoptosis-associated signaling pathways in doxorubicin-resistant cells. Further work will be needed on the discovery of molecules regulating AMPK reciprocally, such as NSC33353 and trastuzumab, as well as on potential methods to enhance doxorubicin sensitivity [175,176].

### 6.3. Metformin

As a first-line hypoglycemic drug, metformin is commonly used to control metabolic disorders and other diseases with abnormal microenvironments, such as cancer. Metformin exerts anti-cancer effects through activating AMPK and has impacts on multiple molecules that are involved in metabolic homeostasis. At the molecular level, metformin activates LKB1 and targets mitochondrial respiratory complex 1, which in turn inhibits the production of ATP and allosterically regulates AMPK without altering its total content [71,177]. The combined treatment of metformin with other drugs, including aspirin and atenolol, demonstrated synergistic effects via AMPK activation and mTOR inhibition [178]. Metformin also increased p53 expression and inhibited DVL3 via the Wnt/β-catenin signaling pathway in MCF-7 and MDA-MB-231 cells [71,179]. Furthermore, the AMPK-dependent apoptotic processes induced by metformin recruited pyruvate kinase M2 (PKM2) [180]. In addition, metformin targets microRNA 27a, which restricts AMPKα2 and alters glucose uptake in cancer cells. In ER-positive cells, the activation of AMPK induced by metformin probably depended on cytochrome P450 3A4 (CYP3A4) conditions [181]. The activation of AMPK downregulates cyclin D1, leading to the release of p27Kip1 and p21Cip1 molecules, and suppresses the cyclin E/CDK2 complex, eventually inducing G0/G1 cell cycle arrest [182]. In MCF7 cells, the activation of AMPK elevated the 5-methylcytosine (5mC) abundance in a thymine DNA glycosylase (TDG) promoter with the help of DNA methyltransferase 3A (DNMT3A), which further suppressed TDG, SREBP1, and ACC1 [27].

The phosphorylation of AMPK and nuclear factor kappa B (NF-κB) induced by metformin is also involved in tumor-associated macrophage polarization and attenuates the release of M2-phenotype cytokines [183]. Metformin blocked the expression of HIF-1α induced by TGF-β in CAFs and promoted the degradation of PD-L1 in the endoplasmic reticulum by phosphorylating the Ser195 site of PD-L1 [146,184]. On the other hand, metformin has the ability to reverse tumorous EMT through AMPK/Akt/MDM2/FOXO3 signaling cascades. Metformin also inhibits the phosphorylation of STAT3 in an AMPK-dependent manner [185]. Metformin inhibits growth-related factors, including the inhibitor of NF-κB α (IκBα), cyclin D1 and ER, but whether these inhibitory effects are involved in the AMPK cascade remains uncertain [186]. The authors of another study stated that the reduction in TGF-β was caused by metformin and that this was beneficial for attenuating EMT [187].

Although the role of AMPK in metformin-induced anti-cancer events has been well elucidated, some of the roles of AMPK under the opposite conditions require further studies. For example, AMPK enhances glycolysis to relieve stress in a pyruvate-dependent manner [188]. Metformin reduces the expression of cell division cycle 42 (CDC42) and AMPK can reverse this effect [189]. It is also worth noting that high levels of AMPK maintained cell survival in dormant ER-positive breast cancer cells [190]. A study showed that combined treatment with metformin and 2-deoxy-D-glucose (2-DG) led to the AMPK activation and the detachment of MDA-MB-231 cells, which suggested cancer-promoting effects of metformin in breast cancer progression [119].

Metformin regulates a set of downstream modulators that have close relationships with AMPK. However, the precise role of metformin in breast cancer treatment, especially in the clinic, is elusive. In some clinical trial reports, metformin reduced FAO in an AMPK-independent manner [191,192]. Furthermore, in a presurgical trial recruiting overweight patients with breast cancer, metformin had no influence on the proliferation of breast cancer cells [193]. Therefore, the network among metformin, AMPK and their downstream molecules mostly depends on the cellular context, and studies of the eventual cell death process triggered by metformin, focusing on aspects such as ferroptosis and the inflammasome, could contribute to a more concise explanation of metformin’s regulation mechanism [194,195].

### 6.4. Tamoxifen

Tamoxifen is a selective estrogen receptor modulator which has been used as an endocrine therapy for patients with ERα-positive breast cancer. Due to the long-term single administration of tamoxifen being likely to cause MDR, concomitant administration is considered as a main strategy to improve clinical outcomes. In tamoxifen-resistant breast cancer cells, overall mitochondrial dysfunctions were observed accompanying AMPK phosphorylation at Ser485/491, but not at Thr172 [196]. However, studies have also shown that tamoxifen regulated AMPK by increasing AMP/ATP ratios and consequently enhancing mitochondrial fusion [11,197]. Interestingly, tamoxifen bidirectionally regulates autophagic processes under various conditions. Tamoxifen blocked the autophagic constituents induced by leptin via the inhibition of ER, but induced autophagy by activating AMPK in MDA-MB-231 cells [155,198]. The sensitivity of tamoxifen was increased in D-glucose deprivation medium, which was probably due to the accumulation of GPER1 caused by AMPK [161]. This demonstrates that in addition to targeting AMPK, tamoxifen may directly act on the downstream molecules of AMPK.

### 6.5. Other Chemical Synthesis Regulators of AMPK

In addition to the common agents mentioned above, some modulators with unique structures exert their anti-breast cancer properties by altering AMPK activity (Figure 3). Liraglutide restricted the expression of microRNA 27a and in turn promoted the expression of the AMPKα2 protein in MCF-7 cells [199]. However, this agent promoted the proliferation of highly invasive MDA-MB-231 cells which possessed AMPK-dependent EMT properties [200]. Alteration of the ATP/AMP ratio is a common mechanism that some modulators apply to activate AMPK [201]. Under glucose starvation stress, oligomycin A was found to induce AMPK phosphorylation in MCF-7 cells. Sodium-glucose co-transporter-2 (SGLT-2) inhibitors which restrict the uptake of glucose also exhibited similar effects [202,203]. In PI3K mutant cells, aspirin was able to inhibit tumor growth by stimulating the AMPK/mTOR signaling pathway [51].

Synthetic chemicals have the potential to affect AMPK levels to attenuate breast cancer, but MDR seems to be a pivotal issue in drug development and in the clinic. Therefore, after the long-term utilization of commercial synthetic drugs, their efficacy in anti-breast cancer should be re-evaluated due to the sophisticated network associated with drug-regulating molecules.

## 7. Potential AMPK Modulators of Natural Products from Herbal Medicines

### 7.1. Berberine

Berberine, which is mostly derived from COPTIDIS RHIZOMA, has multiple anti-cancer properties, including reversing MDR, inhibiting tumorous metastasis, and inducing apoptosis in cancer cells [204]. Previous studies showed that berberine regulates AMPK and targets various downstream pathways in chemo-resistant breast cancer cells. In MCF-7 MDR breast cancer cells, berberine at low concentrations activated the phosphorylation of AMPK and inhibited the expression of HIF-1α and P-gp proteins, while upregulating the expression of the p53 protein at high concentrations [83]. Inversely, another study confirmed that berberine inhibited the phosphorylation of AMPK, as well as downregulating the expression of HIF-1α and P-gp proteins in MCF-7 cells with hypoxia-induced chemoresistance properties [205]. These opposite effects of berberine-regulating AMPK activities were attributed to different environmental stresses and the cellular conditions, such as chemoresistance and the cell phenotypes. To date, the study of the effects of berberine on breast cancer cells has not conclusively determined the regulatory role of berberine in relation to AMPK. Some research has examined the potent effects of gut microbiota and metabolic constituents [206]. In HepG2 hepatoma cells and liver tissues in high-fat-fed mice, berberine attenuated triglyceride accumulation by activating AMPK and in turn decreased the expression of SREBP-1c and stearyl-coenzyme A desaturase 1 (SCD1) [207]. Therefore, although the role of berberine on AMPK in breast cancer is still controversial, we found that the activation of AMPK induced by berberine acts on multiple downstream targets in other disease models.

### 7.2. Curcumin

Several studies have proved the anti-breast cancer efficacy of curcumin, which is rich in the root of CURCUMAE LONGAE RHIZOMA, but few emphasize the necessary role of AMPK. In HCT116 and SW480 human colorectal cancer cells, glucose starvation increased the expression of the neighbor of BRCA1 lncRNA 2 (*NBR2*) via AMPK activation and mTOR inhibition, which formed a feedback loop. Curcumin was able to promote the expression of *NBR2* in HCT116 and SW480 cell lines and further boosted the activation of AMPK [208]. The activation of AMPK induced by curcumin in HepG2 hepatoma cells exhibited interference with PPARα, SREBP-1, and FAS [209]. In H4IIE and Hep3B cells, AMPK activation phosphorylated and inactivated ACC, leading to the inhibition of FASN and the induction of FAO [210]. Obviously, AMPK exhibited the opposite functions in different cell lines. AMPKα blocked the differentiation of adipocytes and inhibited the transcriptional activity of peroxisome proliferator-activated receptor γ (PPARγ) in 3T3-L1 adipocytes. Inversely, AMPKα elevated PPARγ expression in HT-29 cells, which partially described the molecular regulations of curcumin [211]. Moreover, the cell cycle arrest and anti-proliferation events initiated by curcumin-induced AMPKα activation in MCF-7 cells have been attributed to the inhibition of ERK1/2, COX-2 and p38MAPK [211]. In another study on demethoxycurcumin (DMC), one of the major forms of curcuminoids, AMPK was regarded as a monitor for EGFR and heat shock protein 70 (HSP70) in prostate cancer cells [212]. In comparison with curcumin, DMC exerted unique cytotoxicity in MDA-MB-231 cells by activating AMPK and regulating the expression and activities of downstream molecules of AMPK, including mTOR, 4E-BP1, and FASN [213]. Breast cancer patients usually exhibit a high concentration of thrombin, which promotes the expression of cyclin D1 by activating the mTOR and Wnt/β-catenin signaling pathways. Activation of AMPK induced by curcumin in MCF-7 cells was able to reduce mTOR and β-catenin levels and reverse the thrombin-induced proliferation of MCF-7 cells [214]. Curcumin also inhibits the proliferation and invasion of breast cancer via the degradation of Akt. In MDA-MB-231 breast cancer cells, AMPK activation induced by curcumin triggered the autophagy-lysosomal protein degradation pathway, which resulted in Akt protein degradation [215]. The activation of AMPK induced by RL71, another curcumin analog, in TNBC cells has been partially attributed to the inhibition of sarco/endoplasmic reticulum calcium-ATPase 2 (SERCA2) and the promotion of the release of Ca^2+^. This condition initiates the CaMKKβ/AMPK/mTOR signaling pathway [216]. Although a large number of studies on curcumin have accumulated, the applications of curcumin have limitations, especially in regard to drug administration [217]. Therefore, research on nano-formulations, which transform pharmaceutical forms for better bioavailability, is worthy of further exploration.

### 7.3. (−)-Epigallocatechin-3-Gallate (EGCG)

EGCG is the main active ingredient in green tea and inhibits DNA methyltransferase (DNMT) in cancer cells [218]. Although a large number of studies have been carried out on the antitumor effects of EGCG, the efficacy of EGCG in regulating AMPK in breast cancer cells has been less frequently studied. In HT-29 colon cancer cells, EGCG was found to trigger COX-2 inhibition and PGE2 reduction via promoting AMPK expression and phosphorylating ACC. This antitumor effect was mainly based on a dramatic increase in ROS [219]. The activation of AMPK induced by EGCG in HT-29 colon cancer cells also exhibited interaction with VEGF and matrix metalloproteinase-9 (MMP-9) [220]. In addition, AMPK activation is involved in EGCG-induced mTOR inhibition, whereas the suppression of Akt induced by EGCG was found to be irrelevant to AMPK in HT-29 colon cancer cells [221]. EGCG had the potential to hinder protein and lipid synthesis, inhibiting the proliferation of HepG2 and Hep3B cells. EGCG was able to activate AMPK, blocking mTOR, 4E-BP1, and FASN expression, and similar results were observed in H1299 and A549 cells [222,223]. In rat H4IIE cells, EGCG increased the level of split- and hairy-related protein-1 (SHARP-1) partly by regulating AMPK [224]. It is worth noting that EGCG promotes glycolysis and inhibits fatty acid and cholesterol synthesis at the same time. In the case of glycolysis, EGCG activates AMPK through regulating LKB1, and the activation of AMPK subsequently phosphorylates PFK. The inhibition of fatty acid and cholesterol synthesis induced by EGCG is achieved by ACC1 and HMG-CoA reductase (HMGCR) through the LKB1/AMPK signaling pathway [225]. EGCG was also found to inhibit autophagic processes in human melanoma skin A375 cells by inhibiting AMPK phosphorylation [226]. EGCG triggered the demethylase activities of KDM2A and consequently reduced H3K36me2 levels to block rRNA transcription in MCF-7 cells. In this process, the activation of AMPK induced by EGCG is required for KDM2A activation [227]. Hence, AMPK plays an indispensable role in the EGCG-induced inhibition of tumor proliferation, metastasis, and regulation of the microenvironment and immune responses [228,229]. Moreover, the sophisticated signaling network associated with AMPK and other modulators requires further investigation, particularly in relation to AMPK-knockdown tissues.

### 7.4. Ginsenosides

GINSENG RADIX ET RHIZOMA is a precious medicinal herb that is widely used in China and other east Asian countries. Ginsenosides are a series of triterpenoid saponins that are abundant and typical in ginseng. The anti-cancer capacity of ginsenosides has been validated in different experimental models, showing multiple levels of regulation. In BT-474 and T47D cells, ginsenoside-Rg5 treatment activated AMPK and in turn suppressed the activities of p70S6K and S6 proteins [230]. Ginsenoside-Rg2 performed best in inducing cell cycle arrest and apoptosis in MCF-7 cells, as compared with MDA-MB-231 and 293T cells. Ginsenoside-Rg2 also inhibited the phosphorylation of ERK1/2 and Akt and promoted the accumulation of ROS [231]. Furthermore, ginsenoside-Rg2 induced AMPK-triggered mitochondrial dysfunctions by increasing PGC-1α, FOXO1, and isocitrate dehydrogenase 2 (IDH2) mRNA levels [232]. A hydrolyzed ginseng extract GINST was able to activate AMPK and decrease HMGCR expression, which led to the inhibition of cholesterol synthesis [233]. Compound K (C-K, 20-O-(β-D-glucopyranosyl)-20(S)-protopanaxadiol), one metabolite of the ginsenosides, was able to activate AMPK in HepG2 hepatoma cells and relieve lipid accumulation by inhibiting SREBP1c and PPARα [234]. Furthermore, compound K was found to induce autophagy-mediated cell death in A549 and H1975 cells by regulating the AMPK-mTOR axis [235]. 20(S)-ginsenoside Rg3 [20(S)-Rg3] is another extract of ginseng and it induces HT-29 cell apoptosis. Mechanistically, 20(S)-Rg3-induced activation of AMPK decreased Bcl-2 protein expression and increased the levels of p53, Bax, and cytochrome C [236]. There are dozens of ginsenoside derivatives with different antitumor affects. The promotion of AMPK activities is one of their common mechanisms, but more unique signatures should be further identified in breast cancer cells to develop the potential uses of ginsenosides in the clinic.

### 7.5. Paclitaxel

Paclitaxel, which is extracted from TAXUS CHINENSIS, is suitable for HER2-negative and HER2-positive patients and is recommended by the NCCN as a first-line drug in systemic therapies for recurrent unresectable (local or regional) or stage IV (M1) disease. Pharmacological research shows that paclitaxel improves AMPK activity in a dose- and time-dependent manner. The activity of AMPK is crucial for the chemosensitivity of paclitaxel and combined treatment with metforminimproves this chemosensitivity [77]. Elongation factor 1 α (EF1α) is crucial for breast cancer maintenance and acts as a translation factor binding to multiple microtubules in several cancer types, but its expression can be blocked by paclitaxel in an AMPK-dependent manner. Simultaneously, the expression and phosphorylation of FOXO3a, which is triggered by paclitaxel, has a close relationship with AMPK conditions, making the paclitaxel-induced AMPK/EF1α/FOXO3a axis a signature of chemotherapy in breast cancer [237]. A study showed that metformin was able to resensitize MCF7/5-FU and MDA-MB-231 cells to paclitaxel and reverse EMT by activating AMPK [141]. Other studies showed that paclitaxel phosphorylated the Thr172 site of AMPKα and promoted acetyl-Lys379 p53 and SESN2, initiating apoptotic events in MCF-7 and A549 cells [238]. The efficacy of paclitaxel is greatly influenced by the influx transporter, such as solute carrier organic anion transporter family member 1B3 (SLCO1B3), which is one of the main reasons for paclitaxel resistance. AMPK can promote the expression of SLCO1B3 and improve the resistance of paclitaxel in A549 cells. This paradox indicates that some crosstalk exists among paclitaxel, AMPK, and SLCO1B3 [239]. Paclitaxel also exerts anti-breast cancer efficacy by inducing endoplasmic reticulum stress associated apoptosis, whereas autophagy is one of the obstacles in this process. Under endoplasmic reticulum stress, activated p90 ribosomal S6 kinase 2 (RSK2) induced by inositol-requiring enzyme 1α (IRE1α) migrates to the nucleus along with ERK1/2, phosphorylates Thr172 of AMPKα2, and consequently triggers autophagy [240]. Interestingly, pristimerin enhances paclitaxel-associated growth inhibition by inducing ERK-dependent autophagic cell death [241]. Therefore, more approaches to diminishing chemo-resistance in breast cancer could promote the applications of paclitaxel, and the role of AMPK in such regulators could attract interest in the future.

### 7.6. Other Natural Products Targeting AMPK Activity

Through literature mining, we found that many natural products with anti-breast cancer abilities have the function of activating AMPK (Figure 4). 2-DG, a common glucose analog, can dramatically reduce the cellular ATP level, resulting in the activation of AMPK [162]. However, the combined treatment with 2-DG and an AMPK inhibitor exhibited dramatic cytotoxicity in MCF-7 without affecting normal cells, suggesting the presence of other underlying bio-modulators [242]. ω-hydroxyundec-9-enoic acid (ω-HUA), a secondary plant metabolite, activated AMPK in MCF7, MDA-MB-231 and MDA-MB-435 cells by triggering mitochondrial membrane potential deficiency and promoting the production of ROS, which led to apoptosis in breast cancer cells [243]. Baicalein, extracted from SCUTELLARIAE RADIX, could activate AMPK at the Thr172 site and further phosphorylate the Ser555 site of ULK1 in MDA-MB-231 cells, all inducing autophagy-mediated cell death [244,245]. Interestingly, icaritin is generated through the enzymatic hydrolysis of icariin extracted from EPIMEDII FOLIUM, and induces apoptosis in MDA-MB-231 cells by initiating caspase-3 functions, whereas the same molecule was found to stimulate AMPK/ULK1-triggered autophagy in MCF-7 cells, which could be reversed through the use of autophagy inhibitors [91,246,247]. Furthermore, in A549 cells, icaritin inhibited cancer-induced osteoclast generation and promoted apoptotic events via AMPK activation and the subsequent inhibition of IL-6 and TNF-α [248]. 2-hydroxy-6-tridecylbenzoic acid, which is known as ginkgoneolic acid, is extracted from GINKGO FOLIUM and obstructs the viability, migration, and invasion of MDA-MB-231 cells but not of MCF-10A cells [249]. AMPK was activated by 2-hydroxy-6-tridecylbenzoic acid and in turn upregulated the expression of cytochrome P450 1B1 (CYP1B1), as well as reversing the EMT processes in breast cancer cells [249]. Bitter melon extracts were able to promote the phosphorylation of AMPK at the Thr172 site but did not alter the levels of AMPK in either MCF-7 or MDA-MB-231 cells [250]. Furanodiene, isolated from CURCUMAE RHIZOMA, blocked mitochondrial functions and suppressed doxorubicin-resistant MCF-7 cells [251]. Naringenin, which is abundant in citrus fruits and tomatoes, was found to elevate the phosphorylation of AMPKα in E0771 breast cancer cells [252]. Resveratrol, a natural polyphenol, exhibited bidirectional effects on AMPK in MCF-7 and MDA-MB-231 cells by promoting or inhibiting the expression of AMPKα, which could be monitored by the nerve growth factor receptor (NGFR)/AMPK/mTOR pathway, respectively [253,254]. Specifically, the activation of AMPK induced by resveratrol can also be achieved by promoting the expression of SIRT3, and the autophagy triggered by this pathway is considered to be one of the main mechanisms by which resveratrol exerts its effects against cancer [255]. Another advantage of resveratrol is the reduction in liver injury caused by doxorubicin-induced oxidative stress through the AMPK/Nrf2 pathway [256]. Ursolic acid (UA) was reported in a recent study to activate AMPK phosphorylation and in turn inhibit mTOR and PGC-1α, finally impairing mitochondrial functions, which is beneficial to doxorubicin resistance cells [257]. Agents belonging to the statin family are also activators of AMPK and they exhibit potential anti-cancer effects by interfering with the mevalonate signaling pathway. In a combined treatment with LBH589 (HDAC inhibitors), mevastatin phosphorylated LKB1 at the Ser428 site and subsequently activated AMPK in TNBC cells [258]. The same was observed for lovastatin, which is generated from *monascus purpureus* fermentation, and which suppressed glycolysis and induced AMPK activation-associated autophagy in TNBC cells [259]. Moreover, lovastatin-induced activation of LKB1/AMPK increased the expression of p21 and decreased the level of survivin via phosphorylating p38 and p53 [260]. In addition to the modulators described above, vitamin D3, and anisomycin (isolated from *Streptomyces griseolus*) are also regarded as activators of AMPK [261,262].

To date, numerous studies have shown that a great number of natural products can modulate the activity of AMPK and suppress breast cancer cells, but the various downstream signaling molecules triggered by specific natural products make the situation very complex. Hence, further work is warranted to evaluate the safety of these natural products, as well as to examine their extraction and synthetic efficiency in relation to the development of drugs in phytomedicine.

## 8. Conclusions

As a center of metabolic networks, AMPK exhibits regulatory functions in various diseases. Accumulating evidence demonstrates that AMPK plays vital roles in the regulation of human breast cancer. In this review, we found that AMPK is regulated by multiple kinases, genes, and stress conditions. Under these circumstances, AMPK is overexpressed in TNBC cells and inhibited in some other breast cancer cells. The activation or inhibition of AMPK targets various downstream molecules and regulates the growth, proliferation, apoptosis, and MDR of breast cancer cells. Furthermore, we have summarized the pharmacological activators of AMPK, including metformin, berberine, doxorubicin, and AICAR, that activate AMPK and exert anti-cancer activity. We further systematically summarized the natural modulators of AMPK and their mechanism of action in breast cancer cells. The AMPK-related signaling pathways and the mechanisms of AMPK in different contexts in breast cancer cells are complex, and the crosstalk between the activators of AMPK and other pharmacological molecules warrants further exploration.

## Figures and Tables

**Figure 1 molecules-28-00740-f001:**
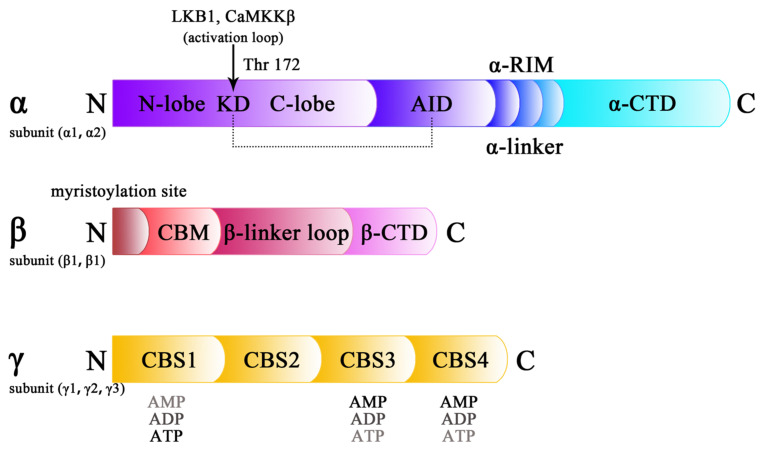
Domain structure and three subunits of the AMPK protein. Upstream kinases, including LKB1 and CaMKKβ, can phosphorylate the activation loop in the kinase domain (KD). Binding of AMP to the γ subunit leads to conformational changes in AMPK, interacts with α-regulatory subunit-interacting motif (α-RIM) and exposes KD to activators, which boost the allosteric activation of AMPK.

**Figure 2 molecules-28-00740-f002:**
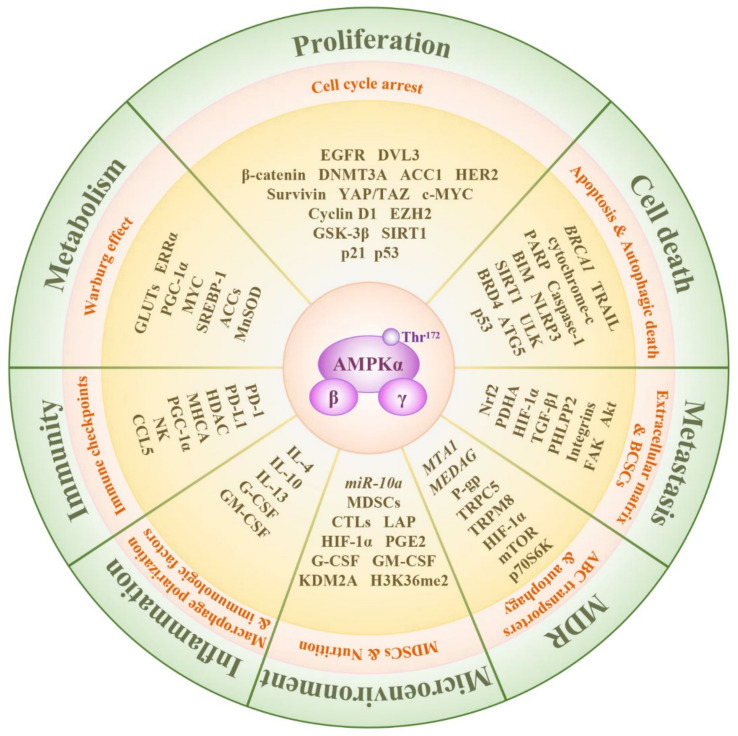
AMPK pathway involved in the oncogenesis of breast cancer. AMPK targets multiple downstream molecules directly and indirectly. In the regulation of proliferation, AMPK blocks the cell cycle and impedes anabolism to inhibit breast cancer cell growth. Apoptosis and the autophagic death of breast cancer cells are two major ways in which AMPK can kill breast cancer cells. In addition, AMPK intervenes in the metastasis, metabolism, and multidrug resistance of breast cancer cells in various ways and plays a role in regulating the breast cancer microenvironment, as well as inflammation and immunity. These processes involve the regulation of the “Warburg effect”, the extracellular matrix, ABC transporters, nutrition, macrophage polarization, and immune checkpoints. It is worth noting that some downstream targets participate in multiple regulatory processes at the same time, and these molecular targets interfere with AMPK and its upstream regulators in these processes.

**Figure 3 molecules-28-00740-f003:**
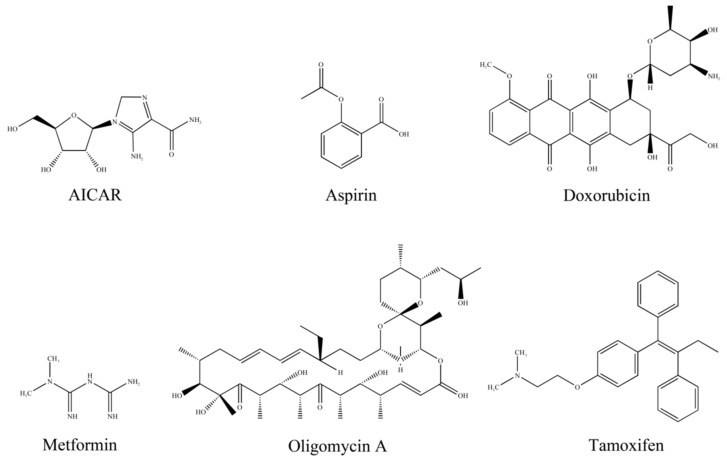
Chemical structures of AMPK chemical synthesis regulators for breast cancer. AICAR: 5-aminoimidazole-4-carboxamide-1-β-D-ribofuranoside.

**Figure 4 molecules-28-00740-f004:**
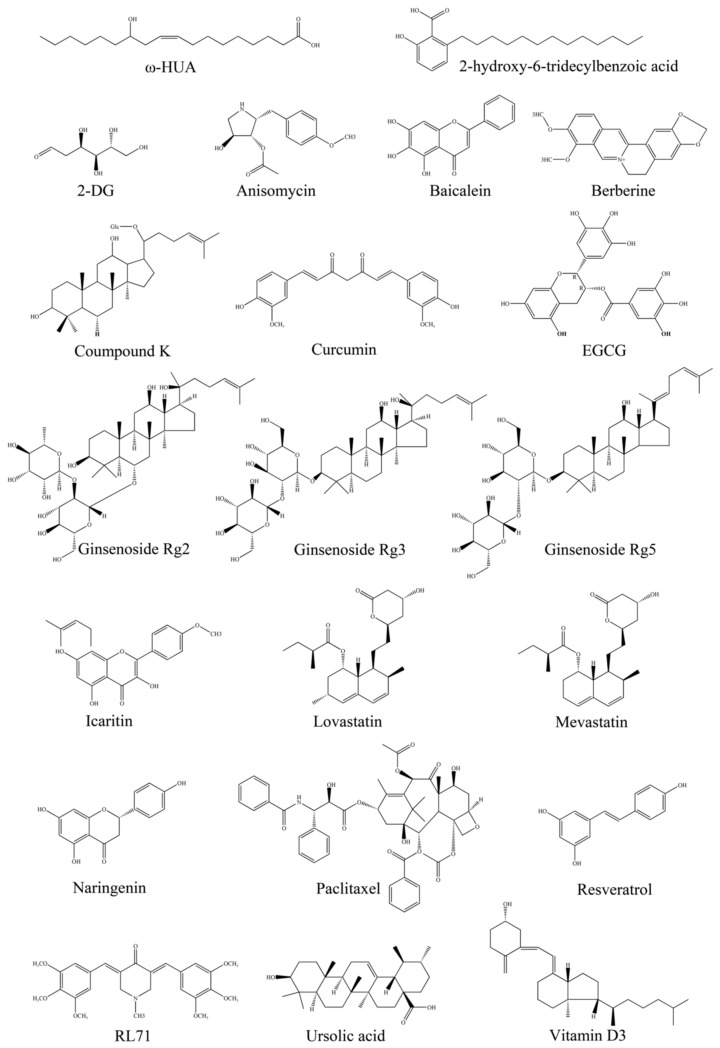
Chemical structures of natural products from herbal medicines used as potential AMPK modulators. ω-HUA: ω-hydroxyundec-9-enoic acid; 2-DG: 2-deoxy-D-glucose; EGCG: (−)-Epigallocatechin-3-gallate; RL71: curcumin analog.

## Data Availability

Not applicable.

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
