# Peer review of "Novel Anti-Cancer Products Targeting AMPK: Natural Herbal Medicine against Breast Cancer"

_molecules, 2023, doi:10.3390/molecules28020740_

Round 1

Reviewer 1 Report

The authors have made a good attempt to provide a comprehensive review of the role of AMPK in cancer progression and treatment. They describe the relevance and structure of AMPK, the expression of AMPK in breast cancer cells and breast cancer related cellular pathways, and finally give a brief description of the known anticancer drugs that employ AMPK activity for the therapeutic effect.

The review can provide crucial insights for future research, and I recommend its publication, after the following revisions:

Minor Recommendations

1.       The title is misleading and unclear. The authors must try to revise and improve to better match the context of the review.

2.       The figure captions are vague. The authors must provide a small description of the figures (especially Figures 1 and 2) to explain and describe the figures.

3.       Figures 3 and 4 are of very poor resolution.

4.        There are several typos throughout the manuscript, particularly in chemical denotations. For example, H2O2 should be written as H2O2.

5.       The authors have missed some recent relevant publications on AMPK activation in breast cancer. Several interesting studies and reviews have been published in 2022 itself, some of which must be included in the references and discussion.

Major Recommendations

1.       The authors talk about the pros and cons of AMPK activation together in the same context, which makes it confusing for the readers to understand the whole point of the discussion. The role of AMPK in breast cancer is ambiguous, with some reports commenting in favor, while some in against. The authors have done a good job of including all the discussion, but it is very confusing and difficult to grasp whether the authors are in favour or against the role of AMPK in breast cancer management. The authors must try to keep the pros and cons distinct (in separate paragraphs?) so readers can easily comprehend the text.

2.       AMPK has also implications for breast cancer stem cell specific pathways. The authors must include a paragraph about that as well since breast cancer stem cells are rapidly emerging as the major source of cancer metastasis and cancer reoccurrence.

Author Response

Response to the reviewers’ comments

23 Now. 2022

Ref. molecules-2036126

Reviewer 1

The authors have made a good attempt to provide a comprehensive review of the role of AMPK in cancer progression and treatment. They describe the relevance and structure of AMPK, the expression of AMPK in breast cancer cells and breast cancer related cellular pathways, and finally give a brief description of the known anticancer drugs that employ AMPK activity for the therapeutic effect.

The review can provide crucial insights for future research, and I recommend its publication, after the following revisions:

Response: Thanks for your positive and professional comments.

Minor Recommendations

  1. The title is misleading and unclear. The authors must try to revise and improve to better match the context of the review.

Response:  Thanks for your professional comments. We have revised the title to better match the context of the review (Title: Novel Anti-cancer Products Targeting AMPK: Natural Inspired from Herbal Medicine towards Breast Cancer).

  1. The figure captions are vague. The authors must provide a small description of the figures (especially Figures 1 and 2) to explain and describe the figures.

Response:  Thanks for pointing out a problem with our work. We have described the main contents of Figure 1 and Figure 2 with explanatory notes. And both figures have been redrawn to ensure better readability.

  1. Figures 3 and 4 are of very poor resolution.

Response: Thank you for your reminder. We rechecked the resolution of each compound image and reformatted all compounds to increase the resolution of the entire image.

  1. There are several typos throughout the manuscript, particularly in chemical denotations. For example, H2O2 should be written as H2O2.

Response: Thank you for pointing out oversights in our work, we have checked and revised the typos throughout the manuscript, including H2O2 and Ca2+.

  1. The authors have missed some recent relevant publications on AMPK activation in breast cancer. Several interesting studies and reviews have been published in 2022 itself, some of which must be included in the references and discussion.

Response: Thank you for pointing out the shortcomings of our work. Based on your suggestions, we have included recent hot literature which includes bioinformatics analysis, AMPK/Nrf2 pathway, lovastatin and AMPK, anisomycin regulating AMPK, EPH receptor A2, Ursolic acid regulating AMPK activity, and Cardiotrophin 1 in tumour microenvironment.

Major Recommendations

  1. The authors talk about the pros and cons of AMPK activation together in the same context, which makes it confusing for the readers to understand the whole point of discussion. The role of AMPK in breast cancer is ambiguous, with some reports commenting in favor, while some in against. The authors have done a good job of including all the discussion, but it is very confusing and difficult to grasp whether the authors are in favour or against the role of AMPK in breast cancer management. The authors must try to keep the pros and cons distinct (in separate paragraphs?) so that the readers can easily comprehend the text.

Response: Thanks for professional comments. To make the readers understand the whole point, we have improved the statement regarding the AMPK role in breast cancer. Generally speaking, in the occurrence of breast cancer cells, the activation of AMPK can exert opposite effects on breast cancer cell proliferation, metastasis, cell death, metabolism and other processes. The inhibition of AMPK is true in cancer cell death under different environmental stresses and the cellular conditions. Therefore, we elaborated the inhibitory and promoting effects of AMPK on breast cancer cells in different status, so that readers can more clearly understand the complexity of AMPK's regulation in breast cancer treatment. For example:

“Previous studies showed that berberine regulates AMPK and targets various down-stream pathways in chemo resistant breast cancer cells. In MCF-7 MDR breast cancer cells, berberine at low concentration activated phosphorylation of AMPK and inhibited expression of HIF-1α and P-gp proteins, while upregulating the expression of p53 protein at high concentration. Inversely, another study confirmed berberine inhibited phosphorylation of AMPK as well as downregulated expression of HIF-1α and P-gp proteins in MCF-7 cells with hypoxia-induced chemoresistance properties. These opposite performances of berberine-regulating AMPK activities are attributed to different environmental stresses and the cellular conditions, such as the chemoresistance and cell phenotypes.”

  1. AMPK has also implications for breast cancer stem cell specific pathways. The authors must include a paragraph about that as well since breast cancer stem cells are rapidly emerging as the major source of cancer metastasis and cancer reoccurrence.

Response: Thank you for your wise criticism. Breast cancer stem cells are an important topic in the occurrence and development of breast cancer, so a paragraph (Section 5.3) has been prepared to highlight the topic, according to your professional suggestions. We totally agree that the content of this part is easier to attract the readers' attention.

Reviewer 2 Report

Bo Peng et al. present a systematic review concerning natural products affecting AMPK activity and their potential application in breast cancer treatment. The topic is of interest; however, the manuscript has several issues the need to be addressed.

Major issues

1.       English needs extensive revision.

2.       The title is difficult to understand. This reviewer understands that the focus of this review are natural products affecting AMPK activity and their potential application in breast cancer treatment. If this is the case, it should be clearly stated.  

3.       Section 3-5. From line 130 to 519 (45% of the manuscript) Authors engage in a lengthy and often confusing description of the role played by AMPK in the pathobiology of breast cancer.  This section should be shortened to 15-20 % of the whole manuscript. See as an example how Hardie and Alessi addressed the role of AMPK in cancer (BMC Biology 2013, 11:36)  

Minor issues

1.       Section 1. The meaning of the sentence in line 72 to 75 is obscure.

2.       Section 2.  In Fig.1 the number of amino acids in each subunit is not correct.   Alpha1 subunits are 559 amino acids long, alpha2 subunits 552. Beta 1 and beta2 subunits are 270 and 272 amino acid long, respectively. Gamma subunits are quite disparate, ranging from 331 amino acid (gamma1) to 569 and 489 (gamma2 and gamma3).  The authors could report these heterogeneities in the text and omit numbers in the figure (see as examples Figure 1 in doi.org/10.1016/j.molcel.2021.08.015; or Figure 1 in doi:10.1038/emm.2016.16).

Section 2. Given the topic of this review, the existence of an allosteric drug and metabolite site at the interface between alpha and beta subunits should be reported (doi.org/10.1016/j.chembiol.2018.03.008).

3.       Section 6 and 7. These sections are the most interesting and informative and should be the focus of the manuscript. But again, English needs to be carefully revised and improved, starting from the title of section 6 (line 522) which is difficult to comprehend.

Author Response

Response to the reviewers’ comments

23 Now. 2022

Ref. molecules-2036126

Reviewer 2

Comments and Suggestions for Authors

Bo Peng et al. present a systematic review concerning natural products affecting AMPK activity and their potential application in breast cancer treatment. The topic is of interest; however, the manuscript has several issues the need to be addressed.

Response: Thanks for your positive and kind comments.

Major issues

  1. English needs extensive revision.

Response: Thanks for your comment. And we have sought help from native English speaker to improve the manuscript and the certificate is attached in the supplementary file.

  1. The title is difficult to understand. This reviewer understands that the focus of this review are natural products affecting AMPK activity and their potential application in breast cancer treatment. If this is the case, it should be clearly stated.  

Response:  Thanks for your professional comments. We have revised the title to better match the context of the review according to your good comprehension (Title: Novel Anti-cancer Products Targeting AMPK: Natural Inspired from Herbal Medicine towards Breast Cancer).

  1. Section 3-5. From line 130 to 519 (45% of the manuscript) Authors engage in a lengthy and often confusing description of the role played by AMPK in the pathobiology of breast cancer.  This section should be shortened to 15-20 % of the whole manuscript. See as an example how Hardie and Alessi addressed the role of AMPK in cancer (BMC Biology 2013, 11:36) 

Response: Thank you for your criticism. Natural products that can regulate AMPK activity are the main content of this review, and the role of AMPK in breast cancer cells and tissues occupies a lot of space to indicate the potential targets for drug discovery. However, according to your suggestion, we also have condensed the content of Section 3-5, reducing the content of this section from the original line 130 to 519 to line 139 to 490, and further enriched the content of the chemical synthesis and natural product sections.

Minor issues

  1. Section 1. The meaning of the sentence in line 72 to 75 is obscure.

Response: We are sorry for confusing you with our work. We checked and found there is indeed an ambiguity in this sentence. In this sentence, what we want to express is that AMPK protein can be regulated by a variety of upstream molecules and act on a variety of downstream targets. Therefore, the types of upstream and downstream molecules contained in different cells affect the pharmacological effects of AMPK. Therefore, this sentence is changed from the “Therefore, the biological functions of AMPK are probably affected by cell contexts that contain sophisticated upstream molecules and downstream targets of AMPK, leading to the controversial potentials in AMPK management of cancer progression.” to “Therefore, as a central regulatory point, activity of AMPK is affected by sophisticated fac-tors in various cancer cell lines and can be modified by pharmacological modulators including metformin, tamoxifen, and some other natural products. Activation or inhibition of AMPK phosphorylation targets multiple downstream molecules and exerts therapeutic functions, leading to the controversial potentials in AMPK management of cancer progression. The regulation of AMPK activity also attenuates other diseases connecting with breast cancer progression, such as inflammation, type 2 diabetes mellitus and obesity.”

  1. Section 2.  In Fig.1 the number of amino acids in each subunit is not correct.   Alpha1 subunits are 559 amino acids long, alpha2 subunits 552. Beta 1 and beta2 subunits are 270 and 272 amino acid long, respectively. Gamma subunits are quite disparate, ranging from 331 amino acid (gamma1) to 569 and 489 (gamma2 and gamma3).  The authors could report these heterogeneities in the text and omit numbers in the figure (see as examples Figure 1 in doi.org/10.1016/j.molcel.2021.08.015; or Figure 1 in doi:10.1038/emm.2016.16).

Response: Thanks for pointing out our error. The references you provided us were of great value, allowing us to explore exactly the number of amino acids contained in AMPK subunits. We retrieved the detailed information of α1, α2, β1, β2, γ1, γ2, γ3 on the website (https://www.uniprot.org/), and perfected this part of the content in this review. We have redrawn figure 1 to make the description of AMPK structure more accurate in this review.

Section 2. Given the topic of this review, the existence of an allosteric drug and metabolite site at the interface between alpha and beta subunits should be reported (doi.org/10.1016/j.chembiol.2018.03.008).

Response: Thank you for pointing out an oversight in our work. The allosteric activation of AMPK is an important part of AMPK activation. We supplement this part of the work in Lines 121-124.

  1. Section 6 and 7. These sections are the most interesting and informative and should be the focus of the manuscript. But again, English needs to be carefully revised and improved, starting from the title of section 6 (line 522) which is difficult to comprehend.

Response: Thank you for your criticism and sorry for the confusion caused by our mistakes in our work. We have re-edited section 6 and section 7 and invited native English speaker to help us correct grammatical errors in the manuscript with the certificate. In addition, we have further enriched these two parts to ensure more complete data marked in red.

Round 2

Reviewer 2 Report

The manuscript was partly improved. However, there are still issues that need to be addressed, starting with the title.

My comments are in the modified pdf file 

English is often poor, 

Citation 9 is a retracted article, this is a serious mistake in a review

Author Response

23 Dec. 2022

Ref. molecules-2036126

To: Prof. Dr. Farid Chemat

Editor-in-Chief

Molecules

Dear Prof. Dr. Farid Chemat

Hope everything goes well with you. We would like to re-submit a manuscript for publication in Molecules, entitled “Novel Anti-cancer Products Targeting AMPK: Natural Herbal Medicine Against Breast Cancer”.

We have carefully revised the manuscript according to the reviewers’ valuable comments, with the help from a native English speaker. It is no doubt that the advice has been very helpful and valuable for improving our present manuscript.

The corresponding modifications also could be found in the revised manuscript with red mark.

Thanks for your careful consideration and the opportunity of revision.

Yours sincerely,

Zhangfeng Zhong & Yitao Wang

[email protected] & [email protected]

Macao Center for Research and Development in Chinese medicine

University of Macau

Avenida da Universidade, Taipa, Macau, China